# Increasing Prevalence and Temporal Trend of Prematurity, São Paulo, Brazil, 2000–2023

**DOI:** 10.3390/epidemiologia6040089

**Published:** 2025-12-08

**Authors:** Adriana Gonçalves de Oliveira, João Batista Francalino da Rocha, Edige Felipe de Sousa Santos, Hugo Macedo Jr., Orivaldo Florencio de Souza, Luiz Carlos de Abreu, Rubens Wajnsztejn

**Affiliations:** 1Programa de Pós-Graduação em Ciências da Saúde, Centro Universitário FMABC, Santo André 09060-650, Brazil; agdeoliveira@gmail.com (A.G.d.O.); joao.rocha@ufac.br (J.B.F.d.R.); hugomacedojr@hotmail.com (H.M.J.); rubens.wajnsztejn@fmabc.br (R.W.); 2Laboratório de Ensino, Pesquisa e Tecnologias em Saúde da Sociedade Brasileira Caminho de Damasco, São Paulo 04581-060, Brazil; 3Departamento de Epidemiologia, Faculdade de Saúde Pública, Universidade de São Paulo, São Paulo 01246-904, Brazil; 4Medical Residency Program in Family and Community Medicine, Centro Universitário FMABC, Santo André 09060-650, Brazil; 5Centro de Ciências da Saúde e do Desporto, Universidade Federal do Acre, Rio Branco 69915-900, Brazil; orivaldo.souza@ufes.br; 6Departamento de Nutrição, Universidade Federal do Espírito Santo, Vitória 29040-090, Brazil; jhgd.marilia@unesp.br

**Keywords:** premature birth, premature newborn, premature labor, spatiotemporal analysis

## Abstract

Background: premature birth is a significant public health problem, especially in developing countries such as Brazil. Premature newborns require special care from birth, often requiring prolonged hospitalization and continuous monitoring by various specialists after discharge. Infant Mortality among children under five years of age in Brazil is alarming, with prematurity being the main cause of death in this age group. Objectives: we aim to analyze the prevalence and temporal trend of premature live births in the state of São Paulo, Brazil, in the period 2000–2023. Methods: this is an Ecological, Time-Series Study with secondary data on premature live births in the state of São Paulo, Brazil, from 2000 to 2023. The variables in this study are aggregated measures by year. The Annual Percentage Change and the Average Annual Percentage Change in the percentage of premature live births were estimated by Joinpoint regression. Results: the percentage of premature live births in the period 2000 to 2023 was 9.71%. The average annual percentage change showed an increase of 2.30% per year. In the 2010 to 2013 segment, there was an increase of 12.58% per year (*p* ≤ 0.05), with subsequent stability in the 2013 to 2016 segment (*p* ≥ 0.05). The largest annual increases occurred in the number of 4 to 6 prenatal consultations and cesarean sections, with an annual percentage change of 4.51% per year and 2.68% per year, respectively. In the birth weight category equal to or greater than 2500 g, there was an increase in premature live births of 2.50% per year. All categories of the variables sex, type of pregnancy, and type of delivery increased in the period 2000–2023 (*p* ≤ 0.05). Conclusions: given the increase in the prematurity rate in the period 2000–2023, a long-term growing trend is expected in the largest and most developed state in Brazil.

## 1. Introduction

Premature birth is a significant public health problem, especially in developing countries such as Brazil. Premature newborns require special care from birth, often requiring prolonged hospitalization and continued monitoring by various specialists after discharge.

Furthermore, infant mortality among children under five in Brazil is alarming, with prematurity being the main cause of death in this age group [1]. Several factors associated with prematurity, in combination with the burden of disease in the population, result in years of life being lost and years lived with disability. Thus, prematurity is currently a serious public health problem [2].

A study conducted in Brazil using a national database of 2.9 million singleton live births in 2015 found that Brazilian municipalities with high cesarean section rates (>80%) had a 21% higher prevalence of preterm births compared to municipalities with low cesarean section rates (<30%), after adjusting for confounding factors related to socioeconomic conditions [3].

Furthermore, the impacts of premature birth are reflected in an increase in deaths in the first five years of life, especially in the most vulnerable regions [1,4]. For children who survive, prematurity increases the risk of chronic conditions, including growth disorders from the neonatal period, delays in motor and cognitive development, and psychological problems, all of which have significant consequences. These children also face an increased risk of developing chronic diseases, such as type II diabetes and cardiovascular disease [5].

Preventing preterm birth is a long-term challenge that is essential to reducing child morbidity and mortality and relieving pressure on health systems. Reducing pre-term birth rates is critical to improving child health and minimizing the impact on health resources. Preterm birth is the leading cause of morbidity and mortality in children under five years of age worldwide, leading to respiratory, infectious, metabolic, and neurodevelopmental sequelae [6]. The true global prevalence of preterm birth has been difficult to determine but was estimated at 9.9% in 2020, with no significant change over the past decade. In the US, this rate was 10.5% in 2021 [7].

Data from the Ministry of Health (MS) show that approximately 340,000 newborns born in Brazil each year are premature. According to the World Health Organization (WHO), in 2023, approximately 10% of births worldwide will be premature. Premature birth, which occurs before 37 weeks of gestation, is responsible for a substantial part of infant mortality, being associated with 35% of neonatal deaths (occurring up to the 28th day of life), in addition to being the second leading cause of mortality in children under five years of age, evidencing its prolonged impact on child health [7].

For premature newborns who survive, the consequences can be severe and long-lasting, affecting multiple body systems and resulting in neurodevelopmental disabilities, chronic respiratory problems, and visual and hearing deficits [2,8].

Thus, prematurity is a “silent emergency” that demands a coordinated global response to improve child health and survival. This includes improving health care, reducing inequalities, and addressing environmental and social factors that contribute to preterm birth [9]. The etiology of preterm birth is multifactorial, requiring special attention to identifiable and modifiable risk factors that can be altered, as well as non-modifiable factors [9,10].

The fragility of premature newborns increases the risk of several sequelae, affecting their development and growth, making it crucial to predict and consider these risks for the implementation of adequate preventive measures [11].

Conducting research in this area is essential to develop effective prevention and intervention measures. Identifying the risk factors associated with preterm birth and developing specific strategies to reduce its incidence will allow us to improve preventive and intervention approaches. This knowledge is crucial to improving neonatal health outcomes and reducing the economic and social impacts associated with prematurity.

Currently, there have been significant advances in the management of prematurity in the state of São Paulo, Brazil. However, significant gaps persist that require attention, especially in research, professional training, and public policies aimed at improving the care and health outcomes of premature newborns. This study fills the gap in up-to-date, representative data on long-term, state-level temporal trends in preterm births. To fill this gap, the objective is to analyze the prevalence and temporal trends of premature live births in the state of São Paulo, Brazil, in the period 2000–2023.

## 2. Materials and Methods

### 2.1. Study Design

This is an ecological, time-series study with secondary data on premature newborns in the state of São Paulo, Brazil, in the period between 2000 and 2023. The units of observation in ecological studies are two or more geographically defined populations (countries or regions) at the same time, or the same population at different points in time [12].

### 2.2. Location

The state of São Paulo is in the Southeast Region of Brazil. In 2022, according to information released by the Brazilian Institute of Geography and Statistics (IBGE) (https://www.ibge.gov.br/cidades-e-estados/sp.html; accessed on 1 January 2025), the population of the State of São Paulo was 44,411,238 people, a population density of 178.92 people per km^2^ and a Human Development Index (HDI) of 0.806 [13].

### 2.3. Data Source

The data source for counting live births and premature live births evaluated in this study was the database of the Department of Information Technology of the Unified Health System (DATASUS) of the Brazilian Ministry of Health (https://datasus.saude.gov.br/informacoes-de-saude-tabnet./; accessed on 30 April 2024) [14].

The DATASUS database stores health information from all locations in Brazil with unrestricted public access. In this study, information was extracted by mother’s place of residence in the state of São Paulo, for the period between 2000–2023.

In the state of São Paulo, the Health Department collects Live Birth Certificates from health establishments or registry offices for home births. Subsequently, information about live births is entered into the Live Birth Information System and incorporated into the DATASUS database.

Data collection was carried out in September 2024. The eligibility criterion was the occurrence of birth with premature newborns and live births in the state of São Paulo in the period 2000–2023.

### 2.4. Study Variables

The variables in this study are aggregated measurements by year, according to [12] who defines in the ecological study the units of analysis as groups of people rather than individuals.

The study variable was defined as preterm live births, which is defined as the duration of gestation up to 36 completed weeks [15]. Prematurity is classified according to the 10th International Classification of Diseases (ICD-10), ref. [16] (P073) for preterm newborn and (O601) for preterm labor with preterm delivery. Information on preterm live births was extracted by year for the total group and stratified for birth sex, type of gestation, mode of delivery, antenatal visits and birth weight.

To assess the quality of the information system, we estimated the proportion of missing/blank data for the “gestation duration” variable for each year. All DATASUS data compilation was performed by two researchers independently, using extraction spreadsheets developed by the authors; a third researcher was responsible for correcting discrepancies.

### 2.5. Statistical Analysis

All information was extracted by the Internet Data Tabulator (TabNet) to a file in Comma-Separate-Values (CSV) format. Subsequently, the information collected was translated and opened in the Microsoft Office Excel spreadsheet to perform calculations of the percentage of premature live births.

The percentage of premature live births was calculated by dividing the number of premature live newborns by the number of live births and, subsequently, the quotient of the division was multiplied by 100. This formula was applied to the total group and stratified by sex, type of pregnancy, type of delivery, prenatal consultations and birth weight.

The annual percentage change (APC) and the average annual percentage change (AAPC) of preterm live birth were estimated by Joinpoint regression to identify the change points in the time series and the trend of each identified segment in the period from 2000 to 2023, with the help of the Joinpoint Regression Program (version 5.2, 2024) [17] developed by the National Cancer Institute, Rockville, MD, USA.

Joinpoint regression models [18] were identified for the total group and stratified by sex, type of pregnancy, type of delivery, prenatal visits and birth weight. In each model, the dependent variable was the percentage of preterm live births.

The independent variable was the year. The options of heteroscedastic errors with constant variation and log-linear models were chosen for the analysis of the models [19].

The Bayesian information criterion (BIC) method was applied to select models. The APC and the AAPC with respective 95% confidence intervals estimated by the parametric method indicated the direction and magnitude of the temporal trends.

Autocorrelation was identified using the Bartlett periodogram test for white noise, using Stata version 17 [20]. A *p*-value of ≤0.05 was used to accept the hypothesis of autocorrelation. Variables with autocorrelation were selected using the first-order autocorrelation option estimated from the data. Joinpoint regression models with a *p*-value of ≤0.05 accepted the hypothesis of annual variation in the period.

### 2.6. Legal and Ethical Aspects of Research

This research was conducted with aggregated secondary data on live births, obtained from the databases of the Ministry of Health published on the internet, in accordance with the current ethical precepts necessary for conducting research with human beings, determined in the Resolution of the National Health Council (CNS) No. 466, of 12 December 2012, ref. [21] and the Law of Access to Information No. 12,527, of 18 November 2011 [22]. It should be noted that this study considered freely accessible data in the information systems and the databases consulted did not include confidential information, such as name and address, so that the approval of the study project by a Research Ethics Committee was waived.

## 3. Results

In the period from 2000 to 2023, the percentage of premature newborns born alive was 9.71%. The male gender had a higher percentage of premature live births. In the type of pregnancy, triplet pregnancy had the highest percentage (91.16%) of prematurely live newborns, followed by twin pregnancy with 55.72%. In the type of delivery, the highest percentage of premature live births identified were those resulting from vaginal delivery (10.17%) in contrast to cesarean delivery (8.86%). In the context of the number of prenatal consultations, there was a higher percentage of premature live births in mothers with 3 or fewer consultations. In relation to birth weight, there was a higher percentage of prematurity in those born alive between 1000 and 1499 g. (Table 1). This section may be divided by subheadings. It should provide a concise and precise description of the experimental results, their interpretation, as well as the experimental conclusions that can be drawn.

The percentage of premature live births per year in the state of São Paulo is shown in Figure 1. In 2002 and 2001, there was the lowest percentage of premature live births, with 7.09% and 7.09%, respectively. In contrast, in 2012, 2013 and 2023, the highest percentages of premature live births occurred, 12.40%, 11.97% and 12.02%, respectively. From 2011 to 2012, there was a significant increase in the percentage of premature live births, followed by a decrease until 2015. From 2017 onwards, there was a constant increase throughout the years in the percentage of premature live births until 2023.

Analysis by annual percentage change (APC) identified four segments of interrupted trends for preterm live births, all with statistical significance: the 2000–2010 period showed an increasing trend (APC: 2.23; *p* = 0.0008); the 2010–2013 period demonstrated an accelerated growth (APC: 12.77; *p* = 0.0008); the 2013–2016 period showed a trend reversal with a decline (APC: −4.75; *p* = 0.0024); and the 2016–2023 period indicated a resumption of growth (APC: 1.78; *p* < 0.0001).

The average annual percentage change (AAPC) confirmed a statistically significant increasing trend for preterm births both in the period prior to the COVID-19 pandemic (2000–2019) (AAPC: 2.60; *p* < 0.0001) and in the period including the pandemic (2000–2023) (AAPC: 2.46; *p* < 0.0001).

By gestational age category, extreme prematurity (20–27 weeks) represents the smallest proportion among premature births, remaining stable, ranging from 4.3% to 6.11% in total. Severe prematurity shows a consistent decline from 10.94% in 2000 to 8.88% in 2023, among premature births. Moderate/late premature birth represents the largest proportion of premature births, with significant growth from 84.04% in 2000 to 85.79% in 2023.

Table 2 shows the average annual percentage variation in premature live births in the state of São Paulo. In the period from 2000 to 2023, in the state of São Paulo, there was a 2.46% increase per year in premature live births. With a higher percentage in male live births (AAPC: 2.53; *p* < 0.0001) compared to female live births (AAPC: 2.38; *p* < 0.0001). Single and twin pregnancies increased with approximate percentage values, being 2.34 per year (*p* < 0.0001) and 2.32 per year (*p* < 0.0001), respectively. There was a greater increase in premature live births by cesarean section (AAPC: 2.68; *p* < 0.0001) in contrast to vaginal delivery (AAPC: 2.00; *p* < 0.0001). The completion of 4 or more prenatal consultations showed a high increase in the average annual percentage variation (*p* < 0.0001). Newborns weighing 2500 g or more had a high increase in prematurity (AAPC: 2.50; *p* < 0.0001). The only category that showed stability in premature live birth was birth weight of 999 g or less (*p*= 0.729) in the segment from 2000 to 2023.

Preterm live births in the female, singleton, and birth weight categories increased, except for the birth weight < 500 g stratum, which decreased, and birth weight 4000 g or more, which remained stable, considering alpha = 0.05. Lower birth weights revealed a higher proportion of preterm infants (Figure 2).

In the preterm infant’s strata by sex, there were four trend segments with APC significantly different from zero at the alpha = 0.05 level: three segments (2000–2010, 2010–2013, and 2016–2023) with an increasing trend and one segment (2013–2016) with a decreasing trend. Both strata showed a long-term increasing trend, with a greater increase between 2000 and 2011 compared to 2012 and 2023.

Preterm live births with birth weights of 2500 to 2999 g had the highest AAPC (2.98; *p* < 0.00001), 95% CI: 2.61; 3.27, with an increasing long-term trend. During this period, the greatest growth occurred in the 2010–2013 segment [APC of 13.57 (95% CI: 7.94; 15.96), *p* = 0.0004] and stability occurred from 2013 to 2023 (*p* = 0.7714).

Preterm infants with birth weight < 500 g showed a long-term decreasing trend (2000–2023), with an AAPC of −0.80 (95% CI: −1.29; −1.19), *p* = 0.0143. In the 2003 to 2006 segment, there was a greater decrease [APC of −13.29 (95% CI: −16.28; −0.51), *p* = 0.0479] and from 2013 to 2016 [APC of −9.17 (95% CI: −11.74; −2.28), *p* = 0.0359]. There was an increase in the segments: 2006 to 2013 [APC of 5.21 (95% CI: 3.33; 9.50), *p* = 0.0395] and 2016 to 2023 [APC of 2.62 (95% CI: 0.88; 5.88), *p* = 0.0275].

Preterm infants with birth weights between 500 and 999 g showed four trend segments between 2000 and 2023: from 2000 to 2004, with an APC of 1.88 (95% CI: 1.36; 3.00), *p* < 0.0001, a decrease from 2004 to 2007 [APC of −1.67 (95% CI: −2.36; −0.59), *p* = 0.0016], a greater increase from 2007 to 2010 [APC of 1.84 (95% CI: 0.88; 2.44), *p* = 0.0012], and remained stable from 2010 to 2023, *p* = 0.1587.

The stratum of premature infants with birth weights of 1000 to 1499 g increased [AAPC of 0.66 (95% CI: 0.58; 0.77), *p* < 0.0001]. From 2000 to 2023, four trend segments were observed: 2000 to 2004, growth [APC of 3.15 (95% CI: 2.51; 4.42), *p* < 0.0001], 2004 to 2007, decline [APC of −2.48 (95% CI: −3.26; −1.15), *p* < 0.0001], 2007 to 2010, resumption of growth [APC of 2.22 (95% CI: 1.15; 2.89), *p* < 0.0001], and, 2010 to 2023, less expressive growth was maintained [APC of 0.29 (95% CI: 0.09; 0.40), *p* = 0.0184].

Preterm infants with birth weights of 1500 to 2499 g increased throughout the period [AAPC of 1.84 (95% CI: 1.62; 2.06), *p* < 0.0001], with the greatest increase in the 2000–2004 segment [APC of 4.29 (95% CI: 3.01; 7.39), *p* < 0.0001]. Those weighing 3000 to 3999 g increased [AAPC of 2.25 (95% CI: 1.25; 3.29), *p* < 0.0001], with significant trend segments from 2010 to 2013, growth [APC of 43.64 (95% CI: 29.24; 51.27), *p* = 0.0171], and, from 2013 to 2016, decrease [APC of −14.40 (95% CI: −19.13; −6.42), *p* = 0.0164]. Preterm infants with birth weights of 4000 g or more showed stability.

Between 2000 and 2023, an inverse relationship was observed between the number of prenatal consultations and the proportion of preterm births. One to three consultations had the highest proportions, ranging from 14.89% to 28.47%, followed by no consultations (15% to 26.86%) and four to six consultations (9.42% to 24.88%). Seven-to-more consultations had the lowest proportions of preterm births (5.05% to 9.18%) (Figure 3).

The behavior of the proportion of preterm births among live births by number of prenatal consultations reinforces the fundamental importance of adequate prenatal care as a strategy for prevention and improvement of perinatal outcomes. It presented distinct time trends: 2000–2008, an increase in rates for all groups and more pronounced growth in the groups with the fewest consultations; 2008–2015, a decrease in measurements was observed in the groups with less prenatal care and seven or more visits; 2015–2023, growth resumed, except in the group with one to three visits, which stabilized (Figure 3).

Pregnancy type was observed to be one of the main determinants of prematurity, with multiple pregnancies representing a very high-risk group that requires specialized obstetric care and intensive neonatal planning. The temporal trend in prematurity revealed a gradual increase in singleton pregnancies from 2000 to 2010; a greater increase occurred from 2010 to 2013; a decrease from 2013 to 2016; and a resumption of growth from 2016 to 2023. Twin pregnancies showed a consistent and significant increase during the period and a gradual deceleration in growth over time. In triplet pregnancies, there was growth from 2000 to 2011, with stabilization from 2011 to 2023 (Figure 3).

The data suggest that the mode of delivery was not an isolated determinant of prematurity, but rather a reflection of maternal and fetal clinical conditions. The increasing trend in prematurity associated with cesarean sections in recent years deserves attention to ensure appropriate indications and timing (Figure 3).

## 4. Discussion

The objective of this study was to analyze the temporal trend of preterm live births in the state of São Paulo, Brazil. We found that the average annual percentage change increased by 2.30% per year from 2000 to 2023. The largest annual increases occurred in the number of four to six prenatal consultations and cesarean sections, with annual percentage changes of 4.51% per year and 2.68% per year, respectively. A statistically significant increase was observed for all categories of the variables sex, type of pregnancy, and type of delivery from 2000 to 2023.

Preterm birth of live babies increased substantially in the state of São Paulo between 2000 and 2023. The largest increase occurred in the number of 4 to 6 prenatal consultations, with an annual percentage change of 4.51% per year. Additionally, the number of cesarean deliveries showed a significant annual percentage increase of 2.68 per year. Surprisingly, premature live births with birth weight equal to or greater than 2500 g showed a temporal trend of increasing by 2.50% per year.

One of the limitations found in the analysis of time series is reported by variables entered in the data collection form. The gestational age (GA) indicator was classified in a grouped manner and without specification of its estimation method. Thus, from 2011 onwards, GA began to be recorded in a disaggregated manner in gestational weeks [23].

In the total number of premature live births and stratified by sex and type of delivery, in addition to premature live births with birth weight equal to or greater than 2500 g, there was a significant increase in the 2010–2013 segment, with a subsequent decline in the 2013–2016 segment. In 2011, there was a change in the content of the Declaration of Live Births, with greater detail in the information collected. As a result, there was, for example, a change in the magnitude of prematurity in the country, which now has a measurement closer to that reported in the Brazilian survey [24]. This fact may have been a factor in the change in the temporal trend of premature live births.

The changes implemented in live birth registries may have contributed to improvements in case reporting and increased efficiency in recording prematurity cases [25]. In addition, there have been changes in health practices, with changes in prenatal care and delivery guidelines, which may have impacted on the way births are recorded [26]. “In addition, socioeconomic and health factors that promoted variations in the population’s living conditions and health may have influenced the fluctuation observed between the periods 2010–2013 and 2013–2016 [27].

These changes may have led to a more systematic recognition of prematurity cases, reflecting not only an epidemiological reality, but also improvements in recording and monitoring practices [28].

Premature live births increased in the period from 2000 to 2023 in single and twin pregnancies. In triple pregnancies, there was an increase only in the 2000–2013 segment, with subsequent stability. It is known that different factors influence the occurrence of prematurity, such as genetic, sociodemographic, [29] environmental [30] and those related to pregnancy [31]. The variations in the type of delivery and type of pregnancy corroborate the findings of other researchers [32,33,34].

There was an increase in premature live births in all categories of prenatal consultations in the period from 2000 to 2023. It is noteworthy that there was an increase in overall assistance to pregnant women throughout these years of study coverage (2000 to 2023). However, this percentage increase was observed, showing that the factors associated with prematurity [32] need to be detailed and based on their identification, tackled one by one with a view to reducing the outcome of prematurity and improving the overall health of the newborn [11].

In the birth weight category equal to or greater than 2500 g there was an increase between 2000 and 2023. Additionally, premature live births increased in the birth weight category of 1500 g to 2499 g and in the category between 1000 g and 1499 g. While it remained stable in premature live births with birth weight equal to or less than 999 g.

A study indicates a decrease in the prevalence of birth weight in 2020 across all weight categories below 2500 g [35]. However, analyzing a single event does not reveal a trend toward decreasing birth weight in premature newborns. A longer-term analysis, from 2000 to 2023, revealed an increase in birth weight in newborns weighing less than 2500 g and more than 1500 g.

It is recognized in the field of public health that prematurity and its complications are responsible for a significant proportion of preventable mortality among newborns. Maintaining the pregnancy until the baby reaches an adequate weight is crucial to reduce the risk of death during early childhood [36,37,38].

The relationship between birth weight and prematurity is significant and multifaceted. It influences cognitive and motor development in childhood and adolescence, as well as health in adulthood [39,40].

Among the various risk factors related to insufficient birth weight is prematurity [36]. The lower the gestational age, the lower the intrauterine weight gain and the lower the birth weight, contributing to neonatal and infant morbidity and mortality [40]. Understanding this relationship is crucial for interventions that aim to improve prenatal care and perinatal outcomes [32,40].

Regarding the sex variable, there were more male live births in the period analyzed from 2000 to 2023. The male sex in premature infants with a corrected age of 20 months is an independent risk factor for worse neurological development [41].

Regarding the worst neurological development outcomes in premature infants, across a wide range of gestational ages, late premature or very premature infants presented lower scores on the mental developmental indexes (MDI) test, at the corrected age of 24 months, and there was a worsening for those of the male sex [42].

Other researchers have also observed that male sex was significantly associated with cognitive delay at 24 months corrected age [43] and was a predictor of worse neurological outcomes at extremely low gestational ages, as demonstrated at 30 months corrected age for male infants born with GA ≤ 25 weeks [44].

Thus, prematurity is a causal factor in the increased vulnerability of newborns. The specific reason for this greater vulnerability of males is still unclear; however, [44] indicate the potential for a relationship with a loss of the adaptive response to prenatal stress and its possible influence on early brain development.

Among other factors associated with the risk of premature birth are those linked to maternal health, level of education and socioeconomic status, and marital status, which, isolated or in combination, contribute to increasing the vulnerability of the pregnant woman and, consequently, of her fetus [32,34].

It is worth noting that the persistence of variables associated with the risk of premature birth prevents a significant reduction in mortality, and factors associated with income inequality, low levels of education and marital issues are those that point to weakness in the support network for pregnant women and influence the results of trends in prematurity rates [37,45].

Income inequality and inadequate housing conditions contribute to the risk of premature birth, as they hinder access to health and care for pregnant women and their babies throughout the gestational period [46]. In addition, poor housing conditions and stress factors, such as violence and lack of care during pregnancy, increase the risk of low birth weight [47].

It is noteworthy that infant mortality is higher among self-identified Black, Asian, and Indigenous populations, compared to white and mixed-race populations. Indeed, prematurity remains more prevalent in this population today [48]. Direct efforts with public policies targeting this population are needed to reduce these rates in the coming years.

In recent years, premature birth rates have remained alarmingly high, highlighting the persistence of preventable causes [36,37]. These data reinforce the complexity of the problem and the need for a comprehensive approach to preventing prematurity, with an emphasis on preventive measures such as preconception care, family planning, regular prenatal consultations, education on nutrition and abstinence from harmful substances that contribute to the reduction of this prematurity outcome [49,50,51,52,53].

Additionally, in the event of prematurity, assistance to the premature newborn must be immediate and adapted to the specific conditions of each case, according to the standards of the Brazilian Society of Pediatrics (SBP) [7,9].

The above highlights the multifactorial nature of premature birth and its consequences for the growth and development of newborns. The vulnerable conditions to which premature newborns are subjected require rapid decision-making by managers and public policy makers, given that the risks and prognoses for this population are imminent and real.

Premature births are influenced by the socioeconomic and health conditions of the place of birth [50]. These are the environmental factors experienced during pregnancy and childbirth that will certainly influence the quality of life in the future, being one of the contributing factors to the high rates of infant mortality, one of the main problems still evident today [48,49].

The aggravating factors are the deficiency of the existing structure for health care and maintenance, the restriction and reduction of technical, human and financial resources, which are not sufficient to meet the demands of families and society, resulting in a lack of adequate technical and specialized monitoring, predisposing pregnant women and their fetuses to risks and harm inherent to human growth and development, both intrauterine and throughout life [45,47,48]. These associated factors show that sociocultural, educational and economic conditions determine the conditions of birth, development, illness and death [15,31,38].

In Brazil, there is a decreasing trend in prematurity, especially in more vulnerable women. However, health services need to improve care for women of advanced maternal age and attract those with few prenatal consultations [28]. In the state of São Paulo, the analysis reports a significant increase in 2012, possibly due to methodological changes.

However, it should be noted that these data are official and come from the Ministry of Health and constitute the best information available for the development of public policies. In addition, the SINASC data in the State of São Paulo presented excellent completeness (≤5%) and, as of 2019, the proportion of missing data was only 0.1%. Another advantage of this study was the possibility of analyzing an extensive database of live births for the largest state in Brazil.

The temporal trend was analyzed in premature live newborns, in the administrative regions of the State of São Paulo, by the number of prenatal consultations, type of pregnancy and type of delivery, as well as by sex and birth weight, with all protocol procedures following the guidelines of the Brazilian Ministry of Health.

Furthermore, in quantitative studies, aggregate measures summarize individual data into a single variable that is representative of a population or group in each context. These aggregations can be made based on different periods or locations and help to simplify and summarize data so that analyses are more feasible and comparable.

This set of information is essential for understanding and assessing the profile of premature newborns over a period, with a view to determining vital risks related to birth conditions, growth and child development, these aspects being components of various health indicators and fundamental for assistance in the maternal and child area.

This approach can be useful for understanding phenomena at a more general level, but it can also introduce the ecological effect, where associations observed in aggregates may not accurately reflect associations in individual data.

Aggregated data have a comprehensive and representative sample, saving financial resources and time, since it is not necessary to collect new data and it is possible to access data covering long periods, allowing analysis of trends and changes over time, as well as frequently including a wide range of variables, which allows exploring multiple aspects of prematurity, such as socioeconomic, geographic and demographic factors.

The standardization of secondary sources allows comparisons with other studies or regions, facilitating the contextualization of results and how data collection was already carried out, allowing researchers to analyze and interpret data regarding the research object. These are strengths that deserve to be highlighted when choosing an ecological study with aggregated data on the population of the state of São Paulo.

Thus, the set of data analyzed demonstrates the need for continuous interventions, directed mainly at pregnant women and solving the possible stressors that induce premature birth, including preconception care, family planning and monitoring of the pregnant woman, as well as collaboration between health institutions, researchers and local communities, essential to improve the results for the mother/newborn binomial. Authors should discuss the results and how they can be interpreted from the perspective of previous studies and of the working hypotheses. The findings and their implications should be discussed in the broadest context possible. Future research directions may also be highlighted.

This study has some limitations, such as the possible occurrences of underreporting of premature births or the existence of blank and ignored variables, associated or not with typing errors and difficulties in filling out the variables, specifically in small municipalities. The strengths of this study are as follows: it is a population-based study that describes the temporal trends of preterm live births in Brazil’s most populous state, with the highest birth rate and an extensive and reliable public database, over a relatively long period of time (24 years), using the most recent reports from the Unified Health System (DATASUS). Furthermore, these data constitute the best information available for developing public policy.

## 5. Conclusions

Prematurity remains a serious public health problem in the state of São Paulo, Brazil. In the period 2000–2023, given the increase in the prematurity rate, a long-term upward trend is expected in the largest and most developed state in Brazil.

There has been an increase in premature birth rates, indicating a growing trend in premature births over the years. Thus, there are changes in maternal and child health conditions, socioeconomic factors and/or changes in health care for the pregnant woman and her unborn child during pregnancy.

Premature birth of live babies increased in the state of São Paulo in the period from 2000 to 2023, being more evident in pregnant women with less than seven prenatal consultations, and there was an increase in premature live births with birth weight equal to or greater than 2500 g in the period.

For the trend in preterm births to stabilize or decline, avoiding the risk of reversal, it is necessary to advance policies that contribute to the continuous monitoring of this outcome, in addition to the importance of adopting and reinforcing intervention measures specifically targeted at the population most affected by this condition.

Further studies are needed to deepen our understanding of the determinants that contribute to the increased prevalence of preterm births and how they interact in a state like São Paulo.

## Figures and Tables

**Figure 1 epidemiologia-06-00089-f001:**
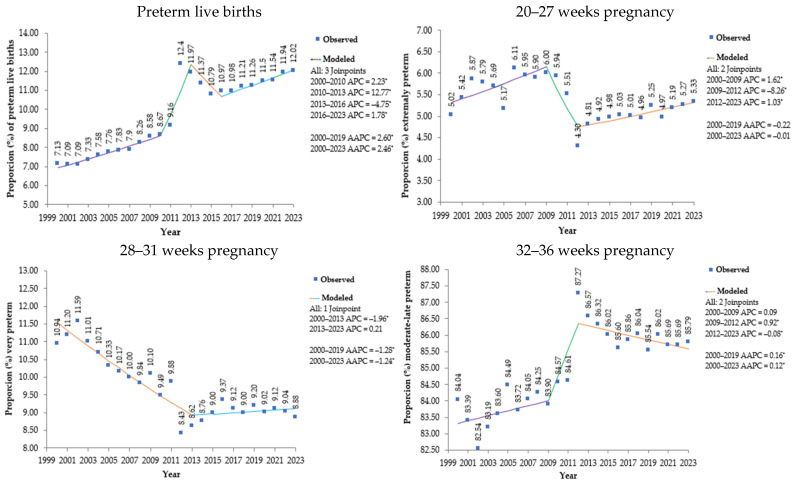
Percentage of premature live births and their distribution by weeks of gestation in the state of São Paulo, 2000 to 2023. Note: * Annual percentage change (APC) and average annual percentage change (AAPC) significantly different from zero at alpha = 0.05 level. Source: prepared by the author based on data from the Department of Information Technology of the Unified Health System (DATASUS) of the Ministry of Health (MS) of Brazil–Information System on Live Births (SINASC), 2024.

**Figure 2 epidemiologia-06-00089-f002:**
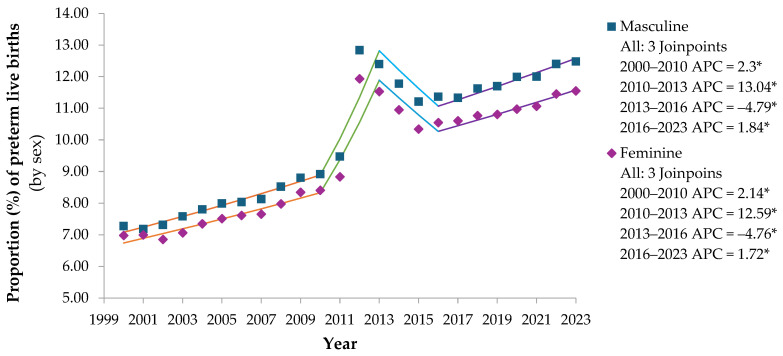
Annual percentage variation in premature live births stratified by sex and by birth weight in the state of São Paulo from 2000 to 2023 (* *p* ≤ 0.05). Note: * Annual percentage change (APC) significantly different from zero at alpha = 0.05 level. Source: prepared by the author based on data from the Department of Information Technology of the Unified Health System (DATASUS) of the Ministry of Health (MS) of Brazil–Live Birth Information System (SINASC), 2024.

**Figure 3 epidemiologia-06-00089-f003:**
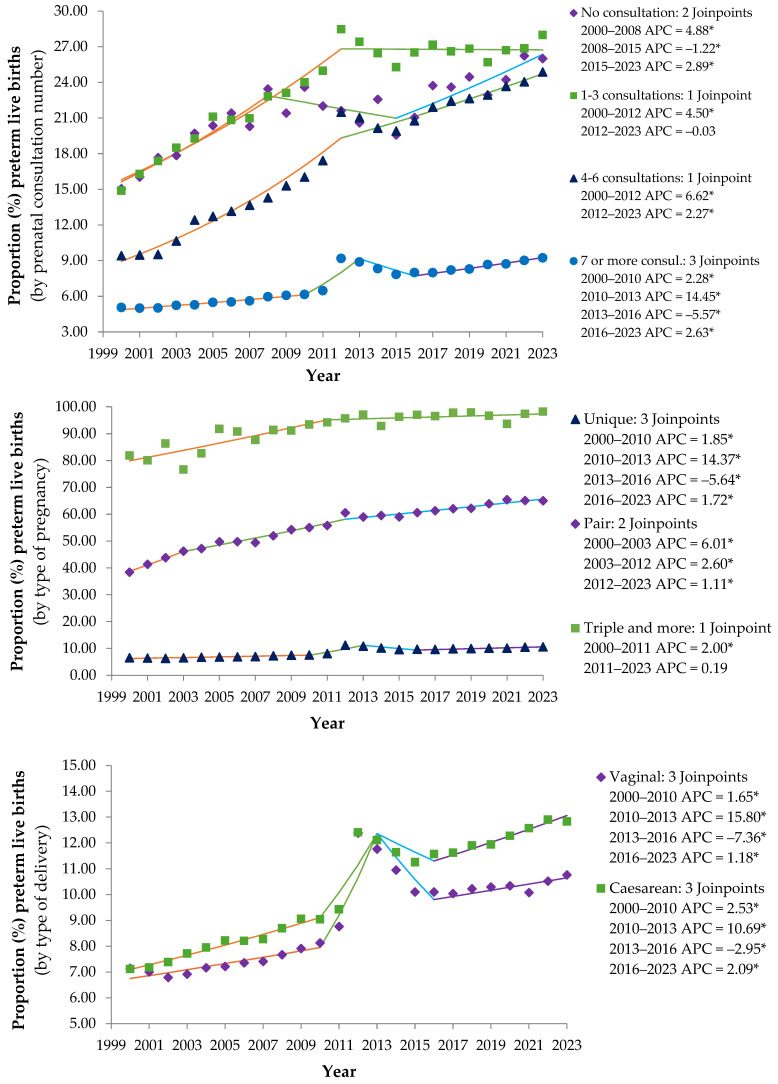
Annual percentage variation in premature live births by prenatal consultation number, type of pregnancy and type of delivery in the state of São Paulo from 2000 to 2023 (* *p* ≤ 0.05). Note: * Annual percentage change (APC) significantly different from zero at alpha = 0.05 level. Source: prepared by the author based on data from the Department of Information Technology of the Unified Health System (DATASUS) of the Ministry of Health (MS) of Brazil–Live Birth Information System (SINASC), 2024.

**Table 1 epidemiologia-06-00089-t001:** Characteristics of premature live births in the state of São Paulo from 2000 to 2023.

Variables	Live Births (n)	Premature Live Births (n)	Premature Live Births (%)
All	14,385,470	1,397,413	9.71
Sex			
Masculine	7,365,827	731,651	9.93
Feminine	7,019,612	651,075	9.27
Type of Pregnancy			
Unique	14,055,225	1,195,200	8.50
Pair	320,290	178,480	55.72
Triple or more	9924	9047	91.16
Type of Delivery			
Caesarean section	6,185,567	548,518	8.86
Vaginal	8,199,872	834,208	10.17
Prenatal Consultation			
None	165,781	34,341	20.71
1 to 3	583,212	133,072	22.81
4 to 6	2,733,607	444,060	16.24
7 or more	10,902,839	771,253	7.07
Birth Weight			
999 g or less	86,960	79,875	91.85
1000 to 1499 g	117,461	108,510	92.37
1500 to 2499 g	1,121,156	574,185	51.21
2500 g or higher	13,059,861	620,156	4.74

Source: prepared by the author based on data from the Department of Information Technology of the Unified Health System (DATASUS) of the Ministry of Health (MS) of Brazil–Live Birth Information System (SINASC), 2024.

**Table 2 epidemiologia-06-00089-t002:** Average annual percentage variation in premature live births in the state of São Paulo from 2000 to 2023.

Variables	AAPC *	(95% CI **)	*p*-Value	Interpretation
All	2.46	(2.26; 2.67)	<0.0001	Growing
Sex				
Masculine	2.53	(2.35; 2.74)	<0.0001	Growing
Feminine	2.38	(1.18; 3.00)	<0.0001	Growing
Type of Pregnancy				
Unique	2.34	(2.13; 2.57)	<0.0001	Growing
Pair	2.32	(2.18; 2.48)	<0.0001	Growing
Triple or more	0.86	(1.00; 1.15)	0.0008	Growing
Type of Delivery				
Caesarean section	2.68	(2.49; 2.92)	<0.0001	Growing
Vaginal	2.00	(1.74; 2.29)	<0.0001	Growing
Prenatal Consultation				
None	2.30	(2.04; 2.58)	<0.0001	Growing
1 to 3	2.31	(2.05; 2.59)	<0.0001	Growing
4 to 6	4.52	(4.13; 4.92)	<0.0001	Growing
7 or more	2.82	(2.61; 3.05)	<0.0001	Growing
Birth Weight				
999 g or less	0.14	(−0.69; 1.00)	0.729	Stability
1000 to 1499 g	0.66	(0.58; 0.77)	<0.0001	Growing
1500 to 2499 g	1.84	(1.62; 2.06)	<0.0001	Growing
2500 g or higher	2.50	(1.40; 3.62)	<0.0001	Growing

Note: * AAPC: Average annual percentage variation. ** CI: Confidence interval. Source: prepared by the author based on data from the Department of Information Technology of the Unified Health System (DATASUS) of the Ministry of Health (MS) of Brazil–Live Birth Information System (SINASC), 2024.

## Data Availability

The data source for counting live births and premature live births evaluated in this study was the database of the Department of Information Technology of the Unified Health System (DATASUS) of the Brazilian Ministry of Health (https://datasus.saude.gov.br/informacoes-de-saude-tabnet/; accessed on 30 April 2024).

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
