# Peer review of "Increasing Prevalence and Temporal Trend of Prematurity, São Paulo, Brazil, 2000–2023"

_epidemiologia, 2025, doi:10.3390/epidemiologia6040089_

Round 1
Reviewer 1 Report
Comments and Suggestions for Authors
The authors present a time-series ecological study examining the prevalence and temporal trends of premature live births in São Paulo, Brazil, from 2000 to 2023. Using national health data and joinpoint regression analysis, the study offers valuable insights into the increasing rates of prematurity and its associations with factors such as type of delivery, number of prenatal consultations, and birth weight. The topic is both relevant and timely, especially considering the global burden of preterm birth. However, several aspects require clarification and improvement, particularly regarding figure formatting and analytical stratification by gestational age.
- The figures presented in the manuscript lack visual consistency and clarity. For instance, Figure 1 does not include a Y-axis label, which hinders interpretability. Additionally, Figure 3 contains axis labels and annotations entirely in Portuguese, whereas the manuscript is written in English. To improve clarity and uniformity, all figures should adopt consistent formatting, especially in terms of axis labels, legends, fonts, and language. I strongly recommend translating all graphical elements into English and ensuring that all figures are fully labeled.
- Although the study defines preterm birth as delivery at ≤36 completed weeks, it is important to acknowledge that clinical outcomes vary substantially across gestational age subgroups. For instance, neonates born at 28 weeks differ markedly from those born at 36 weeks in terms of prognosis and medical needs. I encourage the authors to explore the feasibility of reanalyzing or at least presenting key findings stratified by gestational age ranges (e.g., <30 weeks, 30–33 weeks, 34–36 weeks). This would significantly enhance the clinical relevance and interpretability of the results.
Minor
1. Please ensure all abbreviations are fully defined at first use in the text and tables. For example, "VPMA" should be explained.
2. Several figure captions lack sufficient context or explanation of what is being shown. I recommend expanding the captions to briefly describe the trends illustrated and their significance.
Author Response
Responses to Reviewer 1's Comments.
Thank you very much for taking the time to review this manuscript. We did our best to address the reviewers' suggestions. Please find the detailed responses below and the corresponding revisions/corrections highlighted/tracked in the edits in the resubmitted files.
Point-by-Point Response to Comments and Suggestions for the Authors
Comment 1: The authors present an ecological time-series study examining the prevalence and temporal trends of preterm live births in São Paulo, Brazil, from 2000 to 2023. Using national health data and joinpoint regression analysis, the study offers valuable insights into the rising rates of preterm birth and its associations with factors such as mode of delivery, number of prenatal visits, and birth weight. The topic is relevant and timely, especially considering the global burden of preterm births. However, several aspects need clarification and improvement, particularly regarding figure formatting and analytical stratification by gestational age. The figures presented in the manuscript lack consistency and visual clarity. For example, Figure 1 does not include a Y-axis legend, which hinders interpretability. Furthermore, Figure 3 contains axis legends and annotations entirely in Portuguese, while the manuscript is written in English. To improve clarity and consistency, all figures should adopt consistent formatting, especially in terms of axis legends, captions, fonts, and language. I strongly recommend translating all graphic elements into English and ensuring that all figures are fully captioned.
Response 1: Thank you for pointing this out. We agree with this comment. Therefore, we have created new figures with consistent formatting, especially in terms of axis legends, captions, fonts, and language. Furthermore, we have translated all graphic elements into English and added the new figures to the revised version of the manuscript.
Comment 2: Although the study defines preterm birth as birth at ≤36 completed weeks, it is important to recognize that clinical outcomes vary substantially across gestational age subgroups. For example, neonates born at 28 weeks differ markedly from those born at 36 weeks in terms of prognosis and medical needs. I encourage the authors to explore the feasibility of reanalyzing, or at least presenting, the main findings stratified by gestational age groups (e.g., <30 weeks, 30-33 weeks, 34-36 weeks). This would significantly increase the clinical relevance and interpretability of the results.
Response 2: I agree. According to data from the National Survey on Labor and Delivery, the prevalence of preterm birth was 11.5%, with 74% classified as late preterm (34 to 36 weeks). In our study, the proportion of preterm births between 32 and 36 weeks was >80% during the study period. Although we initially chose to analyze temporal trends in preterm births together (<30, 30-33 = 34 and 36) to provide greater stability to the analyses and reduce random variation and white noise, we made the modifications suggested by reviewer 1 and performed a new analysis of the main findings stratified by gestational age groups (20-27, 28-31, and 32-36) to emphasize this point. This change can be found in Figure 1 of the revised version of the manuscript, located on page 6 in the Results section.
Comment 3: Ensure that all abbreviations are fully defined upon first use in the text and tables. For example, "VPMA" should be explained.
Response 3: We reviewed all abbreviations in the manuscript and defined them upon first use in the text and tables. Additionally, we standardized the use of the terms "Annual Percent Change" (APC) instead of "APV" and "Average Annual Percent Change" (AAPC) instead of "VPMA."
Comment 4: Several figure captions lack sufficient context or explanation of what is being shown. I recommend expanding the captions to briefly describe the illustrated trends and their meaning.
Response 4: We created new figures with consistent formatting, especially in terms of axis labels, legends, fonts, and language, as mentioned previously. Additionally, trend captions have been described as suggested.
Reviewer 2 Report
Comments and Suggestions for Authors
The topic of this manuscript—premature birth trends in São Paulo over a 23-year period—is timely and addresses an important public health issue. However, in its current form, the manuscript requires substantial revision before it can be considered for publication. There are pervasive editorial and formatting errors, unclear statements, and sections that appear to retain template text. Below, I outline major and minor issues, including direct quotations (retaining original line numbers) and suggested improvements.
Comments on the Quality of English Language-
Abstract (lines 22–40)
-
Editorial issues and lowercase sentence starts: Multiple sentences in the Abstract begin with a lowercase letter, e.g.:
“developing countries such as Brazil.” (line 22)
“prolonged hospitalizations and continuous monitoring by various specialists after discharge.” (lines 23–24)
Every new sentence should begin with an uppercase letter. -
Extra spaces: Throughout the Abstract there are additional spaces before and after punctuation, for example between “phys- 22” and “developing” (lines 22–23).
-
Keywords: The Keywords are separated by periods rather than commas (“Premature birth. Premature newborn. Premature labor. Spatiotemporal analysis”), which is incorrect. They should be formatted as “Premature birth, Premature newborn, Premature labor, Spatiotemporal analysis.”
-
-
Citation Style and Reference List (throughout manuscript)
-
The in-text citations appear inconsistent with the typical MDPI/medical style (e.g., lines 53–57 use “(Barros et al., 2018)”, whereas the journal likely requires Vancouver style). Verify and reformat all citations to match MDPI guidelines (e.g., numbered citations in brackets).
-
The reference list must be reorganized and reformatted consistently (font, indentation, punctuation) according to MDPI requirements.
-
-
Introduction – (Lines 53–57)
-
Current text:
“In Brazil, premature birth is related to the rate of cesarean sections in women with 53
higher educational levels, in addition to socioeconomic aspects, such as lack of knowledge 54
of the birth plan, guaranteed as a right of the puerperal woman. In the socioeconomic 55
context, premature rupture of membranes stands out as one of the main factors associated 56
with the outcome of poverty (Barros et al., 2018).” -
It is unclear how higher education correlates with higher cesarean rates and hence prematurity. Please clarify: “How are cesarean rates and educational level interacting to influence prematurity?” If evidence shows Brazil’s cesarean rate is among the highest in the world, explicitly state this and explain the mechanism (e.g., elective cesareans without medical indication).
-
Research gap/innovation: The Introduction does not sufficiently highlight what gap this study fills or why a 2000–2023 time-series in São Paulo is particularly novel. Clearly articulate “what is unknown” and “why this study is needed.” For example, does this study fill the lack of long-term trend data at the state level?
-
-
Materials and Methods – residual template text (Lines 183–189)
-
Quoted template text remains:
“The Materials and Methods should be described with sufficient details to allow 183
others to replicate and build on the published results. Please note that the publication of 184
your manuscript implicates that you must make all materials, data, computer code, and 185
protocols associated with the publication available to readers. Please disclose at the sub- 186
mission stage any restrictions on the availability of materials or information. New meth- 187
ods and protocols should be described in detail while well-established methods can be 188
briefly described and appropriately cited.” -
Required action: Delete these lines in full. Ensure that all relevant details are provided to allow for reproducibility.
-
-
Results Section
-
The Results (e.g., lines 31–38) are presented clearly, with numerical values and annual percentage changes. No major restructure is needed here; however, check for minor spacing errors.
-
-
Discussion – Organization and Clarity (Lines 293–299 and Beyond)
-
Restate objectives and key findings at the beginning: At the start of the Discussion, briefly remind the reader of the study aim and primary finding. For instance: “This study documented a 2.30% average annual increase in premature live births in São Paulo (2000–2023).”
-
In-text template remnants:
“Premature live births increased in the period from 2000 to 2023 in single and twin 293
pregnancies. In triple pregnancies, there was an increase only in the 2000-2013 segment, 294
with subsequent stability. It is known that different factors influence the occurrence of 295
prematurity, such as genetic, sociodemographic (MARTIN, 2017), environmental 296
(HUANG et al., 2018) and those related to pregnancy (ABDEL RAZEQ; KHADER; BA- 297
TIEHA, 2017). The variations that occurred in the type of delivery and type of pregnancy 298
corroborate the findings of PAIVA SILVA et al. (2013), RIBEIRO et al. (2023), RIBEIRO LZ 299
et al. (2023) and PAIVA PINTO et al. (2022).” -
-
Names of cited authors appear in all caps (“MARTIN, 2017”); they should follow standard sentence case (e.g., “Martin et al., 2017”).
-
Many statements merely list previous studies without explaining how their findings compare to the current results. For example, how do those prior prevalence trends align or contrast with São Paulo’s 2.30% annual increase?
-
-
Organization of limitations:
-
Currently, limitations appear scattered (“section is not mandatory but can be added…” at lines 454–455). Delete template suggestions and collect all limitations in a single paragraph near the end of Discussion.
-
-
Recommendation:
-
Reorder paragraphs: (1) summary of main findings, (2) comparison with existing literature (explicitly state similarities/differences), (3) potential explanations (e.g., rising cesarean trends, prenatal care changes), (4) public health implications, (5) study strengths and limitations, and (6) suggestions for future research.
-
-
Eliminate residual template text (Lines 454–455): “section is not mandatory but can be added to the manu- 454
script if the discussion is unusually long or complex.”
Remove these lines entirely.
-
I encourage the authors to resubmit after addressing these major points. .
Author Response
Response to Reviewer 2's Comments.
Thank you very much for taking the time to review this manuscript. Please find the detailed responses below and the corresponding revisions/corrections highlighted/tracked in the edits in the resubmitted files.
Point-by-point response to comments and suggestions for the authors.
Comment 1: Abstract (lines 22–40). Editorial issues and lowercase sentence beginnings: Several sentences in the Abstract begin with a lowercase letter, for example: “developing countries like Brazil” (line 22), “prolonged hospitalizations and continuous monitoring by multiple specialists after discharge” (lines 23–24). Each new sentence should begin with a capital letter.
Response 1: Thank you for pointing this out. We agree with this comment. Therefore, we have revised the abstract and capitalized each new sentence.
Comments 2: Extra Spaces: Throughout the Abstract, there are extra spaces before and after punctuation, for example, between “phys-22” and “developing” (lines 22–23).
Response 2: We removed all extra spaces before and after punctuation.
Comment 3: Keywords: The keywords are separated by periods instead of commas (“Preterm birth. Preterm newborn. Preterm labor. Spatiotemporal analysis”), which is incorrect. They should be formatted as “Preterm birth, Preterm newborn, Preterm labor, Spatiotemporal analysis.”
Response 3: We apologize for this and have made the suggested corrections in the revised version of the manuscript.
Comment 4: The in-text citations appear inconsistent with typical MDPI/medical style (e.g., lines 53–57 use “(Barros et al., 2018),” while the journal likely requires Vancouver style). Please review and reformat all citations to comply with MDPI guidelines (e.g., numbered citations in parentheses).
Response 4: Thank you for pointing this out. We have reformatted all citations to Vancouver style to comply with MDPI guidelines. Additionally, the reference list has been reorganized and reformatted consistently (font, indentation, punctuation) in accordance with MDPI standards.
Comment 5: Current text: In Brazil, preterm birth is associated with the rate of cesarean sections in women with higher education levels, as well as socioeconomic factors, such as lack of knowledge of birth plans, which are guaranteed as a right of postpartum women. In the socioeconomic context, premature rupture of membranes stands out as one of the main factors associated with the outcome of poverty (Barros et al., 2018).
It is unclear how higher education correlates with higher cesarean section rates and, consequently, with prematurity. Please clarify: "How do cesarean section rates and educational level interact to influence prematurity?" If the evidence shows that Brazil's cesarean section rate is among the highest in the world, please state this explicitly and explain the mechanism (e.g., elective cesarean sections without medical indication).
Response 5: One of the inherent limitations of this study, which performed a cross-sectional analysis of the Live Birth Information System database, is that it cannot be used to establish causal inference. The study only identified that women living in municipalities with high cesarean section rates tend to have higher educational levels than those in municipalities with low cesarean section rates. Indeed, the third paragraph was confusing, and we chose to rewrite it according to the suggestions presented by reviewer 2, providing the necessary clarification for this comment.
Comment 6: Research Gap/Innovation: The introduction does not sufficiently highlight what gap this study fills or why a time series from 2000 to 2023 in São Paulo is particularly innovative. Clearly articulate "what is unknown" and "why this study is needed." For example, does this study address the lack of long-term trend data at the state level?
Responses 6: Thank you for pointing this out. We have added the study's gap/innovation to the last paragraph of the Introduction in the revised version of the manuscript.
Comment 7: Quoted template text remains:
Quoted template text remains:
“The Materials and Methods should be described with sufficient detail to allow 183
others to replicate and build on the published results. Please note that the publication of 184
your manuscript implies that you must make all materials, data, computer code, and 185
protocols associated with the publication available to readers. Please disclose at the submission 186
mission stage any restrictions on the availability of materials or information. New methods 187
and protocols should be described in detail, while well-established methods can be 188
briefly described and appropriately cited.”
Required action: Delete these lines completely. Ensure all relevant details are provided to allow reproducibility.
Response 7: We apologize for this and have deleted these lines completely in the revised version of the manuscript. Additionally, we revised the methods section to ensure that all relevant details were provided to allow for reproducibility.
Comment 8: The Results (e.g., lines 31–38) are presented clearly, with numerical values and annual percentage changes. No major restructuring is needed here; however, check for minor spacing errors.
Response 8: Thank you very much for taking the time to review this part of the manuscript. We have removed all extra spaces from the Results section in the new version of the manuscript.
Comment 9: Discussion – Organization and Clarity (Lines 293–299 et seq.).
Restate objectives and main conclusions at the beginning: At the beginning of the Discussion, briefly remind the reader of the study's objective and main conclusion. For example: "This study documented an average annual increase of 2.30% in preterm live births in São Paulo (2000–2023)."
Response 9: We agree with this comment, so we have created a new paragraph at the beginning of the discussion of the new version of the manuscript, briefly addressing the study's objective and main conclusion.
Comment 10: Remains of models in the text:
"Premature live births increased from 2000 to 2023 in
singleton and twin pregnancies. 293 In triplet pregnancies, there was an increase only in the 2000-2013 period, 294
with subsequent stability. It is known that different factors influence the occurrence of
prematurity, such as genetic, sociodemographic (MARTIN, 2017), environmental
(HUANG et al., 2018), and those related to pregnancy (ABDEL RAZEQ; KHADER; BATIEHA, 2017). The variations in the type of delivery and type of pregnancy
corroborate the findings of PAIVA SILVA et al. (2013), RIBEIRO et al. (2023), RIBEIRO LZ
et al. (2023), and PAIVA PINTO et al. (2022).
Cited authors' names appear in capital letters ("MARTIN, 2017"); they should follow the standard capitalization (e.g., "Martin et al., 2017").
Response 10: Thank you for pointing this out. We have reformatted all citations to Vancouver style to match MDPI guidelines, as mentioned previously.
Comment 11: Many statements merely list previous studies without explaining how their findings compare to current results. For example, how do these previous prevalence trends align with or contrast with the 2.30% annual increase in São Paulo?
Response 11: Thank you for pointing this out. We have reworded the discussion section in the revised version of the manuscript to address the reviewer's suggestions.
Comment 12: Organization of limitations:
Currently, the limitations seem scattered ("the section is not required, but may be added..." in lines 454–455). Please delete the template suggestions and Please summarize all limitations in a single paragraph near the end of the Discussion.
Response 12: We added the necessary information about the limitations and reinforced the study's strengths in the last two paragraphs of the Discussion in the revised version of the manuscript. We apologize for this and have deleted these lines entirely in the revised version of the manuscript.
Comment 13: Recommendation:
Reorder the paragraphs: (1) summary of main findings, (2) comparison with existing literature (explicitly state similarities/differences), (3) potential explanations (e.g., increasing trends in cesarean sections, changes in prenatal care), (4) public health implications, (5) strengths and limitations of the study, and (6) suggestions for future research.
Remove the residual text from the template (Lines 454–455): "This section is not required but may be added to the manual-454
script if the Discussion is exceptionally long or complex."
Remove these lines entirely.
I encourage authors to resubmit their manuscript after addressing these key points.
Response 13: We have reworded the discussion section in the revised version of the manuscript to address the reviewer's suggestions, as mentioned previously.
Reviewer 3 Report
Comments and Suggestions for Authors
Reviewer Comments:
Oliveira et al. address a relevant public health issue: the Increase in prematurity in Sao Paulo, Brazil. However, the manuscript requires substantial revision to enhance its clarity, scientific rigor, and structural quality.
- Language and Formatting:
- The manuscript must be rewritten using structured and grammatically correct academic English throughout.
- Capitalization errors are frequent. For example, in the abstract, the word “the” after "Objectives" and "Results" begins with a lowercase “t”. All new sections or sentences should start with capitalized letters.
- Ensure all headings, figure titles, and subheadings follow consistent and professional formatting standards.
- Figures 2 and 3 have subheadings in a non-English language. Please revise them into English for clarity and consistency with the rest of the manuscript.
- Abstract:
- The abstract should follow a structured format: Background, Objectives, Methods, Results, and Conclusions.
- Grammar and capitalization issues must be corrected.
- Content-wise, the abstract should concisely summarize key findings and conclusions without repeating the introduction.
- Introduction:
- The introduction is fragmented and includes multiple short paragraphs with overlapping content.
- Please consolidate these into a coherent narrative that presents:
- The background and importance of the issue,
- A clear rationale for the study,
- The hypothesis and objectives.
- The reader should understand why this study matters and how it contributes to existing literature.
- Figures and Presentation:
- The difference between Figure 1 and Figure 2a is not clear. Authors should explain how each adds unique value or revise to avoid redundancy.
- All figure captions should be self-explanatory and written in proper English.
- Emphasis on Birth Weight:
- While the detailed analysis of birth weight categories is appreciated, the manuscript would benefit from a discussion of epidemiological factors that influence high birth weight—such as maternal BMI, gestational diabetes, prenatal care access, etc.
- Since the authors suggest a relationship between birth weight and preterm birth, a correlation graph per year showing the relationship between these two parameters would be valuable. This would provide strong visual and statistical support for the hypothesis.
- References and Citation Style:
- The reference formatting is inconsistent. In some citations, authors’ names are written in lowercase, and in others in uppercase. Please standardize the reference style across the manuscript according to the journal's guidelines.
- Ensure consistency in punctuation, italicization, and formatting of journal names, volumes, and page numbers.
- Conclusion:
- The current conclusion section largely repeats content from the introduction. This section should instead summarize:
- Key findings of the study,
- Their implications for public health or clinical practice,
- Limitations of the study,
- Potential directions for future research.
- Rewriting the conclusion to reflect actual study outcomes rather than restating the rationale will significantly improve the impact of this section.
Recommendation: Major Revision
The manuscript requires major restructuring and editing before it can be considered for publication. The topic is relevant, and the data appear valuable, but clarity in communication and presentation is essential to realize its full contribution.
Author Response
Response to Reviewer 3's Comments
Language and Formatting:
The manuscript must be rewritten using structured and grammatically correct academic English throughout.
Capitalization errors are frequent. For example, in the abstract, the word "the" after "Objectives" and "Results" begins with a lowercase "t." All new sections or sentences should start with capitalized letters.
Ensure all headings, figure titles, and subheadings follow consistent and professional formatting standards.
Figures 2 and 3 have subheadings in a non-English language. Please revise them into English for clarity and consistency with the rest of the manuscript.
Thank you very much for taking the time to review this manuscript. Please find the detailed responses below and the corresponding revisions/corrections highlighted/tracked in the changes in the resubmitted files.
Point-by-point response to comments and suggestions for the authors.
Comment 1: Abstract:
The abstract should follow a structured format: Background, Objectives, Methods, Results, and Conclusions.
Grammar and capitalization issues must be corrected.
Content-wise, the abstract should concisely summarize key findings and conclusions without repeating the introduction.
Response 1: Thank you for pointing this out. We agree with this comment. Therefore, we have corrected the abstract, maintaining the structured format: Context, Objectives, Methods, Results, and Conclusions in the revised version of the manuscript.
Comment 2: Introduction:
The introduction is fragmented and includes multiple short paragraphs with overlapping content.
Please consolidate these into a coherent narrative that presents:
The background and importance of the issue,
A clear rationale for the study,
The hypothesis and objectives.
The reader should understand why this study matters and how it contributes to existing literature.
Response 2: We apologize for this and have made the suggested corrections in the Introduction, in addition to providing a clear justification for the study in the 12th paragraph of the Introduction in the revised version of the manuscript.
Comment 3: Figures and Presentation:
The difference between Figure 1 and Figure 2a is not clear. Authors should explain how each adds unique value or revise to avoid redundancy.
All figure captions should be self-explanatory and written in proper English.
Responses 3: Thank you for pointing this out. We agree with this comment. Therefore, we created new figures with consistent formatting, especially in terms of axis legends and language. Furthermore, we translated all graphic elements into English and added the new figures to the revised version of the manuscript.
Comment 4: Emphasis on Birth Weight:
While the detailed analysis of birth weight categories is appreciated, the manuscript would benefit from a discussion of epidemiological factors that influence high birth weight—such as maternal BMI, gestational diabetes, prenatal care access, etc.
Since the authors suggest a relationship between birth weight and preterm birth, a correlation graph per year showing the relationship between these two parameters would be valuable. This would provide strong visual and statistical support for the hypothesis.
Response 4: Thank you for pointing this out. However, since the goal is to analyze the prevalence and temporal trends of preterm live births in the state of São Paulo, based on a time-series study with secondary data, we will consider other relevant relationships in future analyses, such as the development of an annual correlation graph showing the relationship between birth weight and preterm birth.
Comment 5: References and Citation Style:
The reference formatting is inconsistent. In some citations, authors' names are written in lowercase, and in others in uppercase. Please standardize the reference style across the manuscript according to the journal's guidelines.
Ensure consistency in punctuation, italicization, and formatting of journal names, volumes, and page numbers.
Response 5: We apologize for this and have reformatted all citations to Vancouver style to comply with MDPI guidelines. Additionally, the reference list has been reorganized and reformatted consistently (punctuation, italics, formatting of journal names, volumes, and page numbers) according to MDPI standards.
Comment 6: Conclusion:
The current conclusion section largely repeats content from the introduction. This section should instead summarize:
Key findings of the study,
Their implications for public health or clinical practice,
Limitations of the study,
Potential directions for future research.
Rewriting the conclusion to reflect actual study outcomes rather than restating the rationale will significantly improve the impact of this section.
Response 6: We agree with this comment; therefore, we rewrote the conclusion and included two paragraphs at the end of the conclusion in the new version of the manuscript, briefly addressing the study's public health implications and possible directions for future research, to address the reviewer's suggestions. Furthermore, the study's limitations are presented in the penultimate paragraph of the discussion section.
Round 2
Reviewer 1 Report
Comments and Suggestions for Authors
The manuscript has been well revised to reflect the reviewer's comments.
Reviewer 2 Report
Comments and Suggestions for Authors
After the revisions, the manuscript has significantly improved in quality and, in my opinion, is suitable for publication.
Reviewer 3 Report
Comments and Suggestions for Authors
Authors responded to the comments appropriately, and it can be accepted now.